



# Water masses and mixing processes in the Southern Caribbean upwelling system off Colombia

Marco Correa-Ramirez[1], Ángel Rodriguez-Santana[2], Constanza Ricaurte-Villota[1], Jorge Paramo[3].

[1]Institute of Marine and Coastal Research (INVEMAR), Santa Marta, Colombia
[2]Departamento F'isica, Universidad de las Palmas de Gran Canaria, España
[3]Grupo de Investigación Ciencia y Tecnología Pesquera Tropical (CITEPT), Universidad del Magdalena, Santa Marta, Colombia

*Correspondence to*: Marco Correa-Ramirez (marco.correa@invemar.org.co)

**Abstract.** The upwelling system off the southern Caribbean coast is probably the main nutrients source that support the
biological productivity in this oligotrophic sea. The Subtropical Water Mass (SUW) that forms the subsurface salinity maximum in the Caribbean is the main source of upwelled waters in this system. Salinity and temperature vs. depth profiles from 4 oceanographic cruises in the western sector of the Caribbean upwelling system, showed that the salinity of upwelled SUW waters is ~ 0.2 g kg$^{-1}$ lower than the SUW salinity in the central Caribbean and have a slight seasonal variation that agree with with the rainy/dry seasons in the region. Besides, the depth of these SUW waters in the continental slope (~100
m) is ~50 m shallower than the SUW depth in the rest of the Caribbean sea. The origin of these modified SUW waters was analyzed using the Mercator numerical model, which reproduces the main vertical characteristics of the subsurface salinity maximum. Modeled data showed that SUW waters upwelled off La Guajira peninsula come from the western Caribbean and arrive to the system transported by an intense Caribbean Coastal Undercurrent (CaCU, mean speed ~ 0.4 m s$^{-1}$). The lower salinity observed in the upwelled SUW waters may be the result of intense vertical mixing processes with diluted surface
waters in the Panama-Colombia gyro that could occur when the CaCU flows below this region before reaching the upwelling zones. The mixing processes in the SUW by double diffusion and mechanical turbulence driven by vertical shear of horizontal currents, were analyzed using the Turner angle and the Thorpe scale, respectively. Double diffusion by salt fingers was found between the SUW and the Central North Atlantic Waters (NACW) with diffusivity values ~ 10$^{-5}$ m$^2$ s$^{-1}$. But the mechanical diffusivity was two orders of magnitude higher (10$^{-3}$ m$^2$ s$^{-1}$) than the double diffusivities in the entire water
column, generating salt fluxes from the SUW towards surface and towards depth over 2 g kg$^{-1}$ m d$^{-1}$. Beyond modifying the salt content in coastal SUW waters, these mixing processes may also alter the nutrient content of upwelling waters, with effects still unknown to the upwelling ecosystem.



## 1. Introducction

Trade winds over the Caribbean Sea generates northward Ekman transport and the upwelling of subsurface waters on the southerm coast of the basin, off Colombia, Venezuela and Trinidad (Andrade and Barton, 2005; Gordon, 1967; Rueda-Roa and Muller-Karger, 2013). This South Caribbean coastal Upwelling System (hereinafter SCUS) is a low latitude tropical upwelling system (~ 10ºN), whose time and spatial variability is determined by a combination of the change in the coast line orientation, the intensification of the trade winds in the Caribbean Low Level Jet (Amador, 2008; Muñoz et al., 2008; Wang,

2007) and the zonal differences in the stratification of the water column (Rueda-Roa and Muller-Karger, 2013). In the SCUS, about 21 upwelling foci have been identified (Castellanos et al., 2002; Paramo et al., 2011), which are distributed in two zones of different upwelling intensity: a Western Zone (WUZ; 74-69.5ºW) characterized by intense winds that generate an intense seasonal upwelling with a high offshore transport of upwelled waters (Andrade and Barton, 2005), an Eastern Zone (EUZ; 70-73ºW) of less-intense but upwelling-favorable across the year winds that generates a lower offshore transport,

where the stratification favors the pumping of colder and deeper waters than upwelled in WUZ (Rueda-Roa and Muller-Karger, 2013). Besides the Ekman transport, the Ekman pumping forced by the trade winds curl also produces a low level upwelling that is only significant in the EUZ because this can not overcome the high thermal stratification in the WUZ (Rueda-Roa and Muller-Karger, 2013).

The nutrient concentration of the upwelled waters in the SCUS increases the phytoplankton productivity (Corredor, 1979; Muller-Karger et al., 2001, 1989), stimulates growth and changes the ecological structure of coral reefs and macroalgae meadows (Diaz-Pulido and Garzon-Ferreira, 2002; Eidens et al., 2014). Estimations suggest that ~ 95% of the small pelagic biomass in the southern Caribbean is sustained by the increase in biological productivity that stimulates the upwelled waters of the SCUS (Rueda-Roa and Muller-Karger, 2013). Upwelled waters in the SCUS come from to the Subtropical Underwater

(SUW) and share the geochemical composition of this water mass (Muller-Karger et al., 2001). It has been suggested that the low nutrient concentration in SUW is responsible for the lower phytoplankton growth observed in the SCUS when compared to the higher phytoplankton production typically observed in the large Eastern Boundary Currents Systems (EBCS: Humbolt, California, Benguela and Canarias) (Corredor, 1979). The SUW originates in the central tropical Atlantic and freely enters into the Caribbean basin through the numerous passages between the Greater and Lesser Antilles (mean sill depth ~ 1,200 m)

(Gordon, 1967), forming a subsurface salinity maximum (SSM) between 100-200 m deep, which extends throughout the Caribbean (Hernández-Guerra and Joyce, 2000). However, early hydrographic profiles developed in the Caribbean have shown that the southern end of the SSM lack of spatial continuity off Colombia and Venezuela coasts, where the SSM is interrupted by the eastward flow of the Caribbean Coastal Undercurrent (CaCU), a possible return flow for the Atlantic waters that have entered in the Caribbean basin (Hernández-Guerra and Joyce, 2000). The waters transported by the CaCU

may have a more recent origin than the SUW waters. It has been suggested that the CaCU originates at surface in the region of the Panama-Colombia gyre (PCG), and progressively deepens as it moves at ~ 0.1 m s$^{-1}$ to the west, reaching a depth of ~



100 m in the Colombian basin and ~ 200 m in the Cariaco basin (Andrade et al., 2003). It is still unknown whether the CaCU is a permanent flow over time or is spatially continuous along the southern Caribbean coast, because the available evidence for this flow comes from isolated hydrographic campaigns performed in different years (Andrade et al., 2003). The

relationship between the CaCU, the SUW and the coastal SSM with upwelled waters in the SCUS remain unclear.

In the large EBCS, coastal undercurrents transport aged waters of equatorial origin, with low oxygen and high nutrient concentration (Chavez and Messié, 2009). Throughout its trajectory beneath the upwelling zones, part of the nutrients transported by the undercurrent waters can be upwelled directly or mixed with shallower waters that are eventually pumped

by the upwelling (Glessmer et al., 2009; Messié et al., 2009). Therefore, the total amount of nutrients that can be supply by upwelling depends on the original nutrients concentration of subsurface waters, on the intensity of the vertical transport generated by the winds (relative to the depth of the Ekman layer) and the mixing processes of nutrients that can occur between water masses in the water column. Some estimations suggest that the turbulent mixing processes transport to surface about 25% of the upwelled nutrients in EBCS, transport that is not reversible during the periods of upwelling relaxation

when the isopycnals descend to their original depth (Hales, 2005). In addition, the higher salt concentration in the subsurface waters compared to the water below them, makes these waters prone to undergo double diffusion processes, which coexist along with turbulent mixing processes in upwelling systems (Arcos-Pulido et al., 2014). In mid latitudes, instabilities generated by double diffusion processes can generate nutrient fluxes similar in magnitude to those induced by mechanical turbulence or mesoscale eddies (Oschlies et al., 2003). The magnitude of these turbulent mixing and double diffusion

processes has not been previously characterized in the SCUS. This information is required to understand the levels of biological productivity observed in the SCUS.

Using observational and modeled data, the present work aims to analyze the origin and vertical structure of the subsurface salinity maximum and its relationship with upwelled waters in the western zone of the SCUS. A first description of diffusive

processes occurring in the water column is made and it is discussed how these processes could affect the nutrients transport in this upwelling system.

## 2. Methods

### 2.1 Observational data

The vertical distribution and the characteristics of the water masses observed in the SCUS were analyzed using CTD (conductivity, temperature, and depth) data from 4 different cruises. Two of these cruises were made at the edge of the continental shelf off Colombia by the National Hydrocarbons Agency (ANH) at the end of the dry season (ANH-I, May-





June) of 2008, and in the rainy season ( ANH-II, November-December) of 2009 (Fig. 1a). A total of 36 stations were made

on the ANH cruises at depths greater than 200 m, using a self-contained Ocean Seven 316Plus profiler equipped with

IDRONAUT full ocean depth, pump-free and long-term stability sensors.



**Figure 1. a) Localization of CTD hydrographic stations. a) The stations of the National Hydrocarbons Agency of Colombia (ANH) were performed during dry season of 2008 (ANH-I, magenta dots) and the rainy season of 2009 (ANH-II, blue dots). b) The Marine Protected Areas project stations (AMPs) were performed at the beginning (AMPs-I, magenta dots) and the seasonal**

**maximum (MPA-II, blue dots) of the upwellig season 2005-2006. Continuous lines between CTD stations represent the position of hydrographic sections shown in figures 3-7 and 10. The edge of the continental shelf is depicted by the 200 m depth isobath (gray line).**

The other two cruises were carried out in the project "Marine Protected Areas (AMPs): a management tool for demersal

fisheries in the northern zone of the Colombian Caribbean" (Universidad del Magdalena - COLCIENCIAS code 020309-

16652). Eighty-six coastal CTD profiles were performed in AMPs cruises within the areas directly affected by the coastal

upwelling on the continental shelf off the Guajira peninsula (Fig. 1b). AMPs cruises were made at the beginning of the

upwelling season of December 2005 (AMPs-I) and during the seasonal peak of upwelling in February 2006 (AMPs-II).



The analysis of water masses was done using graphs of potential temperature versus absolute salinity ($\Theta$-$S_A$ plots). Some CTD profiles were selected to produce depth sections along the coast, which were interpolated with DIVA (Data Interpolating Variational Analysis; Troupin et al., 2012) to produce continuous fields. Satellite information on sea surface temperature and chlorophyll from the MODIS-Aqua mission (https://oceancolor.gsfc.nasa.gov), and winds from the Blended Sea Winds combined product (https://www.ncdc.noaa.gov/data-access/marineocean-data/blended-global/blended-sea-winds),

were also used to establish the start and intensity of the 2005-2006 upwelling season.

## 2.2 Modeled data

In order to analyze the subsurface circulation of the SUW, numerical outputs from the Operational Mercator (Global Ocean Analysis and Forecast System) ocean model were obtained from Copernicus Marine Environment Monitoring Service

(http://marine.copernicus.eu). This model has a spatial resolution of 1/12 degree, starts on December 27-2006, and it produces daily and monthly mean fields of temperature, salinity, currents, sea level, mixed layer depth and ice parameters over the ocean. It also includes hourly mean surface fields for sea level height, temperature and currents. Using the monthly Mercator outputs, the vertical sections in the same location in the ANH-I section were reproduced to compare modeled and cruise data. Horizontal maps at the depth of the subsurface salinity maximum were done to describes the SUW circulation in

the Caribbean.

## 2.3 Diapycnal mixing parameters

The diffusion of properties generated by turbulent mixing is several orders of magnitude more efficient than molecular diffusion (Thorpe, 2005). Double diffusion by salt fingers is one of the most important processes of turbulent mixing in the central waters of the tropical and subtropical oceans (Schmitt, 1981). The susceptibility of the water column to vertical

convection through double diffusion processes can be evaluated with the Stability Ratio $R\rho$ (Thorpe, 2005a),

$$R\rho = \alpha\,\partial\Theta / \beta\,\partial S_A \tag{1}$$

and the Turner angle (Ruddick, 1983; Turner, 1973),

$$Tu = \tan^{-1}\left(\alpha^\Theta \frac{\partial\Theta}{\partial z} + \beta^\Theta \frac{\partial S_A}{\partial z}, \alpha^\Theta \frac{\partial\Theta}{\partial z} - \beta^\Theta \frac{\partial S_A}{\partial z}\right) \tag{2}$$

where $\alpha$ is the coefficient of thermal expansion, $\beta$ is the saline contraction coefficient, $\Theta$ is the conservative temperature and $S_A$ is absolute salinity. The Turner Angle ($Tu$) is an estimate that allows to determine the relative influence of temperature and salinity in the stratification of the water column and the probability of double diffusion processes. The first of the two

arguments of the arctangent function is the "y"-argument and the second one the "x"-argument in a Cartesian plane. The Turner angle Tu is quoted in degrees of rotation. If -45º < $Tu$ < 45º, the water column is statically stable. If -90º < $Tu$ < -45º, the column is unstable and prone to presenting double diffusion convection of diffusive type. If 45º < $Tu$ < 90º, the column is





unstable and is propense to develop a double-diffusion convection in the form of salt fingers, with greater activity close to 90º. The salt diffusivity in a salt finger favorable regime can be estimated by:

$$K_{sf} = \frac{K^*}{1+(R\rho/Rc)^n} + K^\infty \quad , \tag{3}$$

where  $K^* = 10^{-4} \, \mathrm{m^2 s^{-1}}$  is the maximum diapycnal diffusivity associated with salt fingers,  $K^\infty = 3 \times 10^{-5} \, \mathrm{m^2 s^{-1}}$  is the diapycnal diffusion constant due to processes unrelated to double diffusion, like the internal wave breaking;  $Rc = 1.6$  is the threshold of density ratio where the mixing due to salt fingers falls sharply due to the absence of staircases; n = 6 is an index to control the $K_{sf}$ rate decay with the increase of $R\rho$.

CTD measurements can also provide valuable information of the mixing induced by vertical shear of the horizontal flow throughout Thorpe scale, which is an estimates of the vertical overturning scale associated with the large eddies of turbulence in an otherwise stably stratified fluid (Thorpe, 2005). To quantify the Thorpe scale, density profiles are sorted to produce an ideal stable state, keeping track of the minimum vertical distance each parcel of water had to be moved to establish the stable condition. The Thorpe displacement assigned to depth $d_i$ is

$$L_i(d_i) = d_f - d_i \quad , \tag{4}$$

where $d_f$ is the depth to which the point originally at $d_i$ has been moved. Thus, positive (negative) Thorpe displacements correspond to downward (upward) relocation of a water parcel. The Thorpe length scale $L_T$ is the root-mean-square of all Thorpe displacements within a complete overturn cell, defined as a vertical distance over which Thorpe displacements sum to zero. Questionable instabilities were detected and removed from $L_T$ profiles using an overturn verification trough the overturn ratio:

$$R_o = min(L^+/L, L^-/L) \quad , \tag{5}$$

where $L$ is the total vertical extent of an overturn cell and $L^+$ ($L^-$) is the cumulative extent occupied by positive (negative) Thorpe displacements within the cell.

The errors associated with the determination of salinity from CTD measurements temperature, conductivity and pressure are generally propagated into the derived density profiles as noise and spike-like anomalies, which biases $L_T$ estimations. To 160  eliminate these, speaks in $S_A$ and $\Theta$ profiles were removed before the density calculation, excluding all values whose absolute increase from their previous value was greater than 3 standard deviations of all the increases in the profile. Thus, in the case of salinity, the excluded values corresponded to the depths where  $|\Delta S_{A(i)}| > 3 * std(\Delta S_A)$  , where $\Delta S_A$ is defined by

$$\Delta S_{A(i)} = S_{A(i)} - S_{A(i-1)} \quad . \tag{6}$$



The remaining noise in the density profiles was eliminated by constructing an intermediate density profile, as is clearly
explained in (Gargett and Garner, 2008).

Salt diffusivity associated with shear turbulence $K_{sT}$ is calculated using the Osborn parameterization, as $K_T = 0.128 L_T^2 N$
where $N$ is the Brünt-Vaisala frequency. Salt diffusive fluxes caused by both salt fingers and shear turbulence, were
calculated following the Fick's law as

$$F_x = K_x * \partial S_A / \partial z \quad , \tag{7}$$

where the subscript $x$ could take the form of $sf$ or $T$ depending on the case.

## 3. Results

### 3.1 Water Masses and mixing processes in the upwelling system

The $\Theta$-$S_A$ plot of CTD data from ANH cruises (Fig. 2) shows the presence of 4 water masses off the Colombian coast. On
surface (over first 50 m) most of $S_A$ and $\Theta$ values are in the range of the typical values of the Caribbean Surface Water (CSW,
$S_A = 35$ g kg$^{-1}$, $\Theta = 29$ ºC), a water mass formed in the Caribbean by by a mixture of North Atlantic Surface Waters with
riverine waters incoming in the basin from the Orinoco and Amazon rivers, plus the direct contributions of rain and fresh
water from the South American rivers that are discharging within the basin (Cherubin and Richardson, 2007; Wüst, 1964).
However, the data of ANH-II cruise carried out during the rainy season of 2009, show surface waters more diluted than CSW
towards west of 76.5ºW, with salinity values <33 g kg$^{-1}$ (gray dots, Fig. 2). This low salinity is the result of a greater dilution
in the PCG region caused by large rivers discharging into the Gulf of Darien (Beier et al., 2017).

Under the CSW, higher salinity waters are observed with values close to the typical ones reported for Subtropical Under
Water (SUW, $S_A = 37$, $\Theta = 22$ºC, Fig. 2), forming the subsurface salinity maximum. These waters have ~ 0.2 g kg$^{-1}$ less salt
relative to the SUW in the central Caribbean and show a slight variation in salinity (~ 0.1 g kg$^{-1}$) between the dry season
(ANH-I, May-June 2008) and the rainy season (ANH-II, November-December 2009). This difference could suggest a
seasonal intensification in the mixing processes and a greater dilution of the SUW waters with the less saline surface waters
during the rainy season.



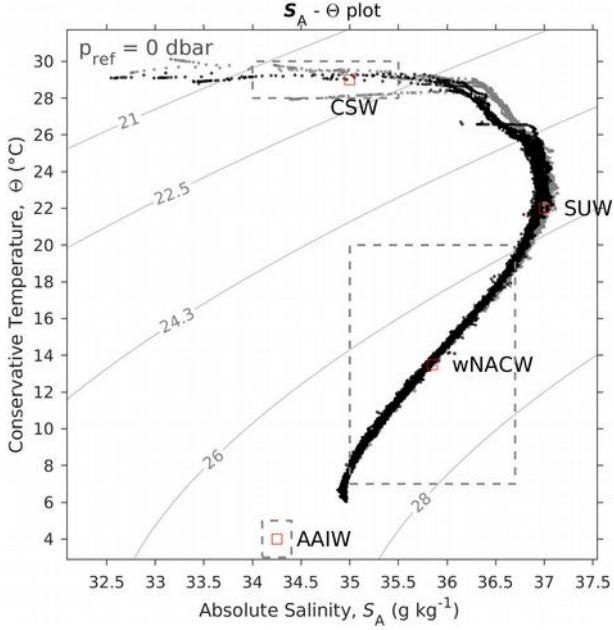

**Figure 2. The conservative temperature – absolute salinity (Θ-$S_A$) plot of the CTD data from ANH-I (black dots) and ANH-II (gray dots) cruises. Red squares represent the typical values for the Caribbean Surface Water (CSW), the Subtropical UnderWater (SUW), the western North Atlantic Central Water (wNACW) and the Antarctic Intermediate Water (AAIW).**

In the center of the Caribbean basin the Subsurface Salinity Maximum (SSM) of SUW waters is located around 150 m depth (Hernández-Guerra and Joyce, 2000). In the ANH hydrographic sections front the continental shelf of Colombia the SSM is observed 50 m shallower (~ 100 m depth) than the center of the Caribbean (Fig 3 c and d). During the dry season (ANH-I) the SSM reaches the surface east of 73ºW front the Guajira Peninsula (Fig. 3c). This upwelling of saline and colder SUW waters may be linked to the seasonal intensification of upwelling favorable winds that occurs during the dry season (Andrade and Barton, 2005). During the rainy season (ANH-II), the SSM is slightly deeper than what was observed in the dry season, although it is still observed the progressive lifting of the SSM towards the east. Unfortunately, the ANH-II section (Fig. 3d) did not have hydrographic profiles in front off La Guajira, so it was not possible to observe the depth of the SSM during this period of upwelling relaxation.






**Figure 3.** Hydrographic sections along the coastal shelf off Colombia made from selected CTD stations of cruises ANH-I (left panels) and ANH-II (right panels); (a-b) Conservative temperature ($\Theta$); (c-d) Absolute Salinity ($S_A$); (e-f) Salt diffusivity by salt fingers ($K_{sf}$, color tones) and Turner angle (black contours); (g-h) salt flux by salt fingers ($F_{sf}$). Dashed lines in panels c-h are the contour of $S_A$=36.8 g kg$^{-1}$ showing the depth of the subsurface salinity maximum zone.





The high salt concentration of the SUW waters at the SSM depth creates favorable conditions for the double diffusion
210    processes and the formation of salt fingers with the less saline and colder waters located below the SSM (Schmitt, 2005;
Schmitt et al., 1987). The vertical mixing of salt by double diffusion could be one of the processes that determine the lowest
salinity observed in the SUW waters off Colombia. Below the SSM, the $S_A$ and $\Theta$ of the water decrease to the typical values
of the wNACW ( $S_A$ = 35.85 g kg$^{-1}$, $\Theta$ = 13ºC, Fig. 2) around 300 m depth, and at 600 meters depth these values  fall to $S_A$ =
35 g kg$^{-1}$ and $\Theta$ = 6ºC, approaching the typical values of the AAIW ( $S_A$ = 34.25 g kg$^{-1}$, $\Theta$ = 4ºC). The Turner angles below
the SSM depth ranges between 45 and 90 °, which shows a high probability of salt finger formation in the entire water
column below 100 m depth. However, the higher salt diffusivities for the salt fingers ($K_{sf}$ > 4x10$^{-5}$ s$^{-1}$) are observed between
300-500 m depth where $Tu$> 75º (Fig. 3e-f), suggesting that the double diffusion processes are faster below wNACW waters.
Despite this, the vertical salt gradient below the SSM makes the negative (towards the depth) salt flux produced by the salt
fingers more intense between 250-350 m depth ($F_{sf}$<-2 x10$^{-2}$ g kg$^{-1}$ m d$^{-1}$) suggesting that the largest double diffusive salt
transport occurs between SUW waters and WNACW waters (Fig. 3g-h).

The mechanical turbulence produced by the vertical shear of the horizontal currents is another process that can potentially
contribute to the mixing of the SUW waters with the adjacent water masses above and below the depth of the SSM. The
stratification in the water column generally precludes the occurrence of this type of mixing process. The Brünt-Vaisala
frequency ($N^2$) shows the stratification of the Caribbean front Colombia is dominated by salinity, which makes the water
column highly stable to the depth of the SSM (Fig. 4c-d). The entry of lower salinity waters during the rainy season (ANH-
II) further increases the surface stratification in the first 50 m depth (Fig. 4d). This makes the mechanical turbulence
diffusivity ($K_{sT}$) is higher below the SSM depth than in the surface layer and generates a high salt flux ($F_T$> 2 g kg$^{-1}$ m d$^{-1}$)
from the SSM towards waters at 300 m depth (Fig. 4G-h) which is ~ 2 orders of magnitude greater than the flux generated by
the salt fingers. Although the $K_{sT}$ values at surface are lower than the deep ones, the intense salt gradient generated by the
entry into the surface layer of low salinity waters during the rainy season also generates a positive salt flux from the SSM to
the surface, which can achieve comparable values to those observed in depth (Fig. 4h). Therefore, mechanical mixing is
observed as the dominant process of salt diffusion from SUW waters towards waters above and below the SSM.

The net flux of salt generated by the mechanical turbulence and the salt fingers depends on the total time during which the
SUW waters have been subjected to these mixing processes and the intensity of the currents at the depth of the SSM. Since
not currents measurements were made on the ANH cruises, the circulation pattern of the SUW in the Caribbean was
analyzed through the numerical outputs of the Mercator model.





**Figure 4.** Hydrographic sections along the coastal shelf off Colombia made from selected CTD stations of cruises ANH-I (left panels) and ANH-II (right panels); (a-b) zoom to the surface 300 m depth of the absolute salinity section ($S_A$); (c-d) Brünt-Vaisala frequency ($N^2$); (e-f) salt diffusivity associated with shear turbulence ($K_{sT}$); (g-h) mechanical salt flux ($F_{sT}$). Dashed lines in all panels are the contour of $S_A$=36.8 g kg$^{-1}$ showing the depth of the subsurface salinity maximum zone.



## 3.2 Subsurface circulation at the salinity maximun depth in the Caribean

Figure 5 reproduces the hydrographic section of the ANH-I cruise using the numerical outputs of the Mercator model, for the
months corresponding to the dry season (ANH-I) and the rainy season (ANH-II). This section shows that the model is
capable of reproducing the main hydrographic characteristics observed in the ANH sections; A surface layer (0-50 m) with
low salinity to the west of 75ºW in the PCG region, a subsurface salinity maximum located ~ 100 m, and waters with
characteristics similar to wNACW waters below 600 m depth. However, there is a slight difference in the halocline of the

Mercator profiles, which is around the isohaline depth of $S_A$ = 36.2. The modeled halocline is narrower and 50 m less deep (~
200 m) with respect to the halocline observed in the ANH profiles, located at ~ 250 m depth. This determines an SSM layer
with less vertical amplitude in the Mercator sections. This difference could be linked to an inadequate modeling of the
diffusive processes in the water column, mainly those associated of double diffusion by the salt fingers, which produces an
underestimation of the salt diffusivity towards the depth from the MSE. The double diffusion parameterization is currently

one of the major challenges of ocean circulation models, as it significantly affects the estimates of the speed of large-scale
thermohaline circulation (Zhang et al., 1998) and the efficiency of vertical nutrients transport (Dietze et al., 2004).

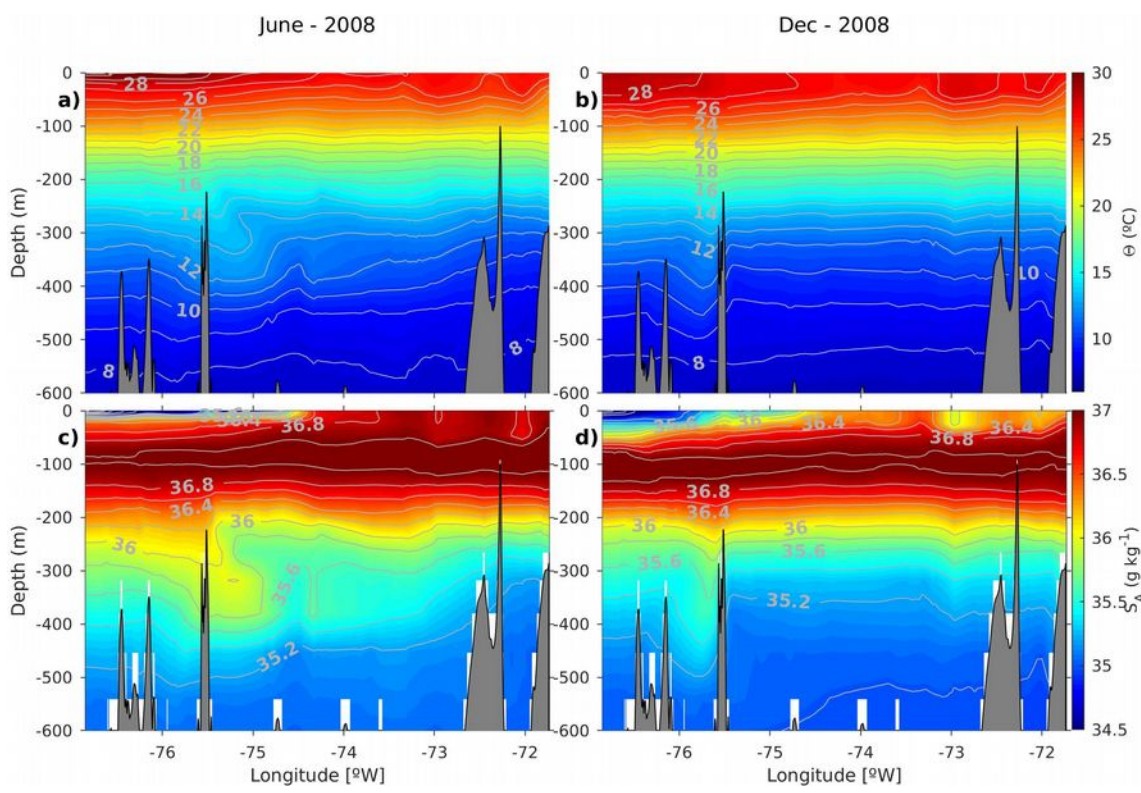

**Figure 5. Hydrographic sections along the coastal shelf off Colombia made from the numerical outputs of the Mercator
operational model for the dates June (left panels) and december (right panels) of 2008. Modeled sections were done in the the same
location of the ANH-I section. (a-b) Conservative temperature (Θ); (c-d) Absolute Salinity ($S_A$).**



Fig. 6 shows the monthly variability of salinity and currents modeled by Mercator in a section along 74ºW from Colombia (14.5ºN, Tayrona National Park, Santa Marta) to Haiti (18 ° N, Port-Salut). Throughout the year, the offshore currents (north of 12ºN) are observed flowing predominantly towards the west (negative values in blue tones, left panels in Fig. 6). A
mesoscale variability of currents is also observed, with narrow sectors (~ 1 ° of amplitude) of strong currents flowing to the east that are limited by sectors of similar size flowing to the west, a characteristic circulation pattern of mesoscale eddies often observed in transit throughout the Caribbean, structures that represent the main variability source of the currents in the basin (Centurioni and Niiler, 2003; Jouanno et al., 2008; Richardson, 2005).

On the slope of the Colombian continental shelf (south of 12ºN), an intense westward flow is observed during most of the year, forming a core at ~ 100 m deep with velocities of ~ 0.4 m s$^{-1}$. The position of this subsurface flow corresponds to the location of the Caribbean Coastal Undercurrent (CaCU) reported by Andrade et al. (2003). However, the intensity of the CaCU reproduced by the Mercator model is 4 times higher than that reported by Andrade et al. (2003). The modeled currents also suggest that the flow of the CaCU is not permanent throughout the year. Its greatest intensification occurs between the
months of January to April, then weakens during the month of May and disappears almost completely in June. In the months of July and August the CaCU intensifies again, but disappears during the months of September and October. Finally, between the months of November to December a new pulse of intensification - weakening of the CaCU is observed.

The intensification periods of the CaCU observed in Fig. 6 correspond to the seasonal intensification of the coastal upwelling
in the western zone of the SCUS. Off Colombian coast the upwelling intensifies mainly in two periods; between the months of December to March and in the month of July, in response to the intensification of the trade winds in the Caribbean low level jet (Andrade & Barton, 2005). During the periods of intense upwelling (January - April and July - August) the modeled salinity sections (Panels on the right side, Fig. 6) show that the waters of the SSM rise to the surface near the Colombian coast, to be subsequently advected towards the north beyond 13ºN. The model thus reproduces the upwelling of subsurface
waters from the SSM, similar to that observed in the ANH profiles, and suggests that the intensification of the CaCU could have a dynamically link with the intensification of the upwelling. Since upwelled waters come from the same depth of the CaCU, most of them must been replaced by waters transported by the CaCU, which could induce the observed acceleration of the CaCU to supply new subsurface waters to the upwelling system. This could also suggest that a significant portion of the SUW waters upwelled in the system would not come directly from the center of the Caribbean. Instead of this, upwelled
waters could come from coastal subsurface waters that are transported from the west of the Caribbean by the CaCU.





**Figure 6. Mercator model sections on a transect along 74ºW, from Tayrona National Park (Colombia) to Port-Salut (Haití). Left panels; monthly means of zonal currents where blue (red) tones represent westward (eastward) velocities. Right panels; monthly means of salinity for the same year.**

Figure 7 shows how the depth and salt concentration of the modeled MSE vary spatially in the Caribbean. In the north of the basin, the SSM is between 120 and 140 m deep and has a higher salt concentration than in the south, with $S_A$ between 37.0 and 37.1 g kg$^{-1}$ (Fig. 5a-b) that are near the original salinity of the SUW in the North Atlantic. In the southwest sector of the Caribbean, the $S_A$ of the MSE decreases to 36.9 and its depth becomes more superficial. During the season of intense





upwelling (from January to March) the cyclonic circulation in the PCG contributes to raise SSM, with the formation of two domes of ~ 100 m depth between 84°-75°W (Fig. 7c). In front of the coasts affected by the upwelling (Peninsula of la Guajira and the Venezuelan coast) the SSM reaches its shallowest depth in the whole basin at ~ 70 m deep. When the upwelling weakens during October-December the SSM deepens again in the southern Caribbean to ~ 100 m depth (Fig. 7d).


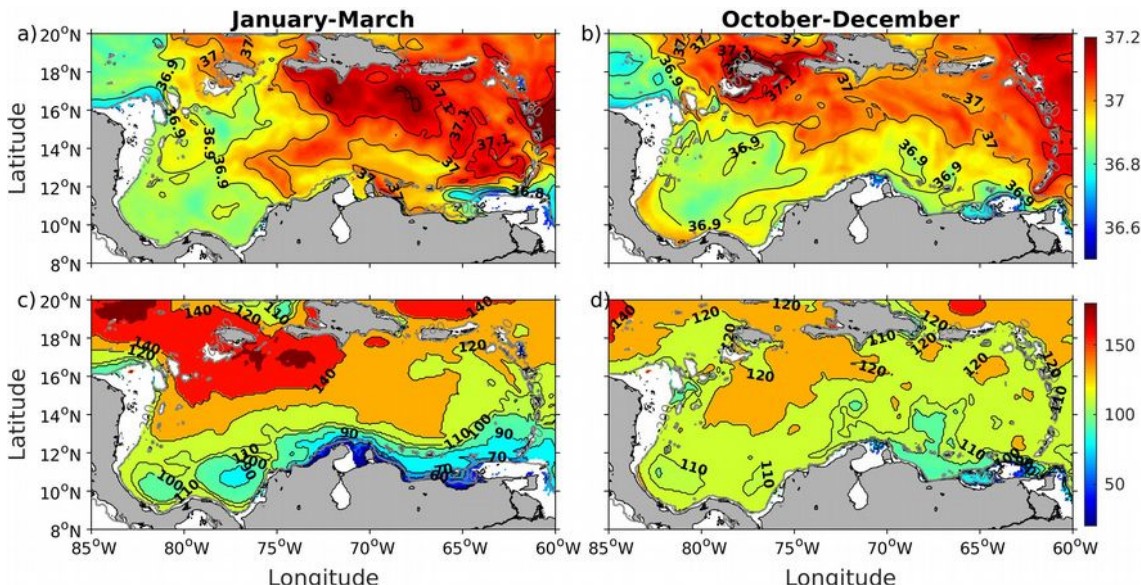

**Figure 7. Salinity (a, b) and depth (c, d) of the Subsurface Salinity Maximum (SSM) in the Caribbean simulated by the Mercator model for the periods of intensification (January-March) and relaxation (October-December) of the upwelling of 2008.**

The currents in the depth of the SSM presented in fig. 8 allow to observe the circulation of the SUW in the Caribbean basin.
This figure shows that the SUW waters are entering the basin through the passages between the Greater and Lesser Antilles, which agree with previously reported by Gordon (1967). After entering, SUW waters flow predominantly to the west, from the Venezuela basin to the continental shelf of Nicaragua and the Jamaican ridge. In some sectors the currents of the SSM are inverted towards the east, due to the presence of mesoscale eddies, structures that can modify the kinetic energy in the water column up to 800 m of depth (Jouanno et al. al., 2008). The SUW waters that flow into the Nicaragua shelf divide around
83°W and 12°N, generating two branches that flow along the platform, one flowing to the northeast and the other flowing to the southwest. The northeast branch surrounds the continental shelf of Nicaragua and then exits towards the Cayman basin through the passages of the Jamaica ridge. The southwest branch changes its direction when it reaches the continental shelf off Costa Rica and continues eastward along the shelfs of Panama, Colombia and Venezuela, forming the CaCU. Therefore, the currents simulated by the Mercator model suggest that the CaCU is formed front the Nicaragua platform, from the
divergence of the subsurface water flow in the SSM depth. This differs from what was suggested previously by Andrade et al. (2003), who propose that the CaCU have a more recent formation from the deepening of surface waters in the PCG





region. The transport of SUW waters by the CaCC from Nicaragua could be connecting at the subsurface level the high dilution region of the PCG (Beier et al., 2017) with the upwelling zones off Colombia and Venezuela, which could explain the lower salinity observed in upwelled waters.


**Fig 8. Subsurface currents at the depth of the Subsurface Salinity Maximum (SSM) in the Caribbean for the periods of intensification (January-March) and relaxation (October-December) of the upwelling of 2008. Blue (red) tones represent westward (eastward) velocities**

The simulated currents also show that the intensity of the CaCU varies seasonally and even loses its spatial continuity during some periods of the year. During the intense upwelling season (Fig. 8a), cyclonic circulation intensifies in the PCG region, which may contribute to the SSM somerization observed in the Fig. 7c. During this period, the CaCU is more intense in the



PCG region and in the Guajira peninsula (70-73ºW) with speeds between 3-4 m s⁻¹ (Fig. 7a), but it is less intense in Venezuela (~ 2 m s⁻¹) and completely absent between Cabo de la Vela (71.6ºW) and Cabo San Roman (70ºW), which
suggests that the CaCU does not have a continuous flow from Colombia to Venezuela during this time. During the upwelling relaxation season (Fig. 8b), the CaCU weakens in the Guajira Peninsula, flows continuously between Colombia and Venezuela, and intensifies slightly in Venezuela, reaching speeds of up to 2.5 m s⁻¹.

### 3.3 The 2005-2006 upwelling season off Colombia coast

Since the ANH sections provided information on the water column only at the edge of the continental shelf, the variability of
the water column inside the continental shelf was analyzed using the coastal CTD profiles made during the AMPs cruises (Fig. 1b). These profiles were made off the Guajira Peninsula at two different times during the 2005-2006 upwelling season (Fig. 9). During this season, Ekman offshore transport calculated from the winds along the coast, began to increase in mid-October from -1 m² s⁻² (not favorable to upwelling) to reach values of 7 m² s⁻² (favorable to upwelling). The cruise AMPs-I takes place in mid-December, near the onset of the season (Fig. 9a). The upwelling of the cold subsurface waters generated
by the increased Ekman transport gradually decreases the surface temperature from 30 to 26ºC during this initial stage. Between December 2005 and February 2006, the Ekman transport remained favorable to upwelling (~ 4 m² s⁻²), showing several intensification impulses with a synoptic periodicity between 5-7 days. During this period, the surface temperature fall to 24ºC, which was the lowest temperature of the season. The cruise AMPs-II was carried out in mid-February 2006 in the days before the maximum (minimum) of the Ekman transport (surface temperature), so this campaign corresponds to the
seasonal maximum of the upwelling. The satellite images of the sea surface temperature show how in the initial stage of the upwelling season (December) the low temperature waters were restricted to the first 100 km of the coastal zone (Fig. 9b), and later in the seasonal maximum (February) the upwelled waters were advected offshore beyond the 14º N forming a tongue of cold water (Fig. 9d).




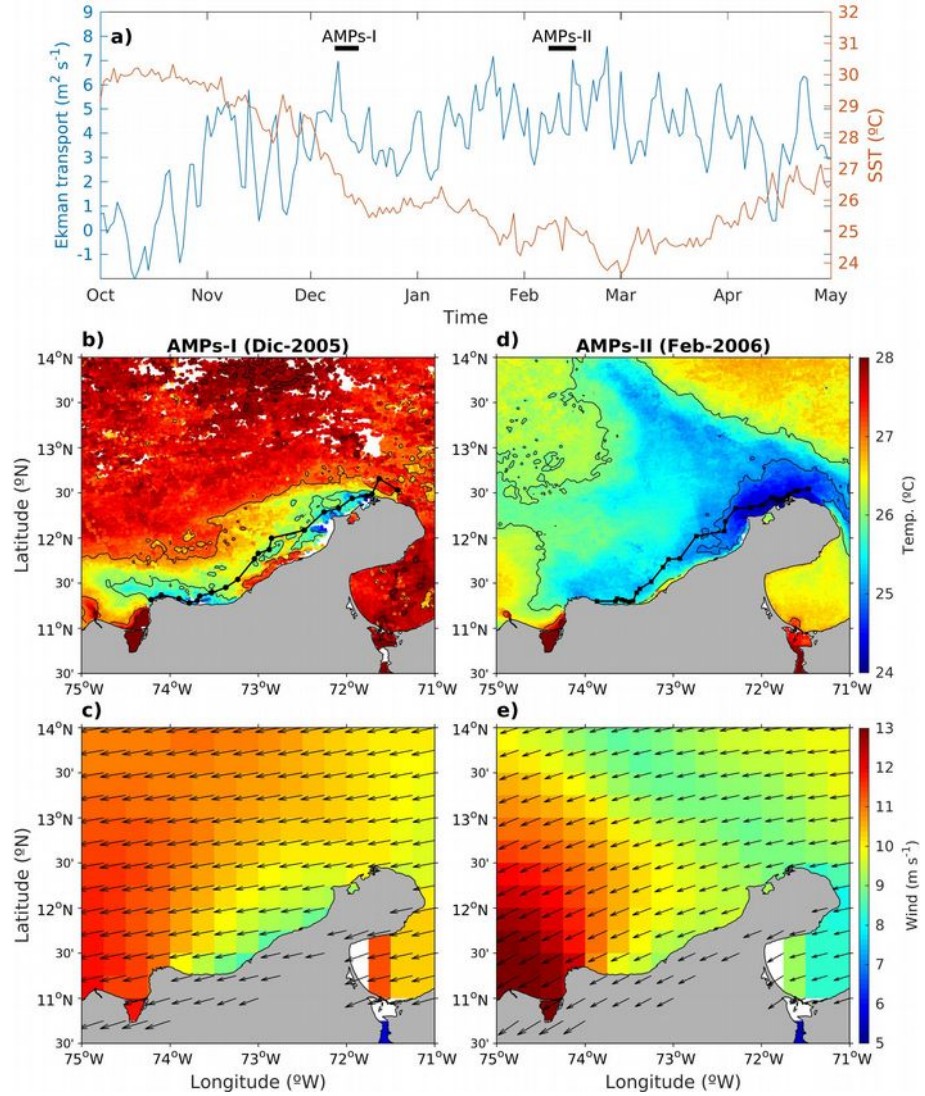

**Figure 9. a) Variability of the Ekman transport derived from the satellite alongshore winds (blue line) an the sea surface temperature off the Guajira peninsula during the upwelling season 2005-2006. Horizontal black bars shown the dates of AMPs-I and AMPs-II cruises. b) and c) shown the mean sea surface temperature and the mean winds during AMPs-I cruise (December 2005). d) and e) shown the same fields but for AMPs-II cruises. Black dots and lines in b) and d) shown the location of CTD stations and the hydrographic sections of Figure 10.**

In the AMPs-I section at the beginning of the upwelling season, the SSM is observed between 30 and 70 m depth (Fig. 10b), a lesser depth than observed in the ANH cruises (~ 100 m depth). The projections of trees (3) of SUW waters of high salinity and low temperature from the depth of the SSM to the surface (approximately 73.8ºW, 72.9ºW and 72.2ºW) show the places where upwelling were active during this time (Fig. 10A -b). In contrast, on surface to the east of 71.8ºW, the high



temperature and low salinity waters produce a high stratification of the water column (Fig. 10e), which shows that upwelling has not yet started in the north of the Peninsula of Guajira.

**Figure 10. Hydrographic sections on continental shelf off the Guajira peninsula made with selected CTD stations from AMPs-I (December 2005, right panels) and AMPs-II (February 2006, left panels) cruises. (a-b) conservative temperature (Θ), (c-d) absolute salinity ($S_A$); (e-f) Brünt-Vaisala frequency ($N^2$).**

During the seasonal maximum of upwelling (AMPs-II), the SSM layer expands vertically and occupies almost the entire water column from ~ 120 m depth to the surface (Fig. 10d). The Θ-$S_A$ diagram shows that most of the surface and subsurface waters correspond to the diluted SUW waters (with ~ 0.2 g kg$^{-1}$ less of salinity, Fig. 11) observed in the ANH sections. These





waters may have been pulled over the coastal shelf forced by the upwelling. The highest temperature observed in these SUW

waters could have been obtained by reaching the surface or by mixing with warmer surface waters. In fact, the Brünt-Vaisala

frequency during this time showed a lower stratification compared to the start of the upwelling season, which could favor an

intense vertical mixing in the water column.

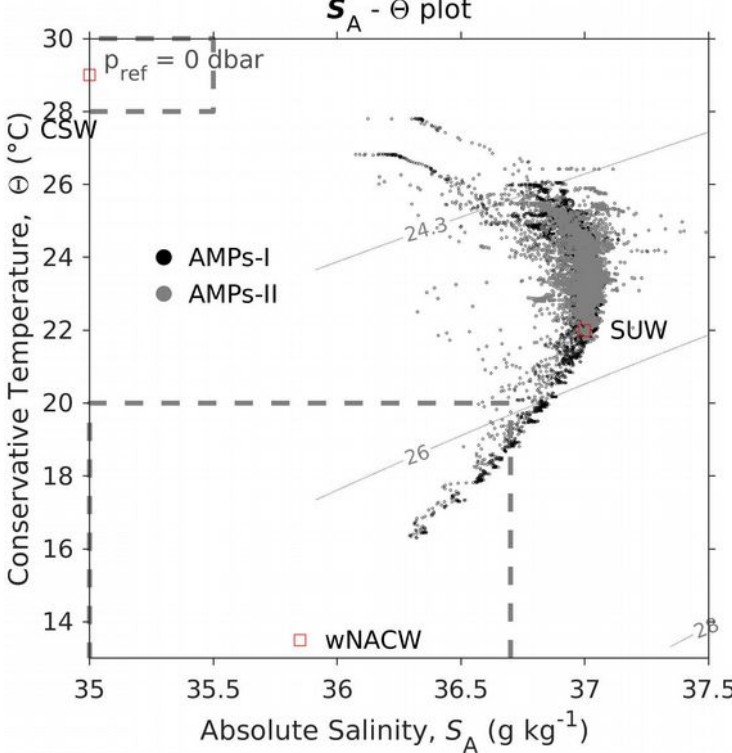

**Figure 11. The conservative temperature – absolute salinity (Θ-$S_A$) plot of the CTD data from AMPs-I (black dots) and AMPs-II (gray dots) cruises. Red squares represent the typical values for the Caribbean Surface Water (CSW), the Subtropical UnderWater (SUW), the western North Atlantic Central Water (wNACW).**

## 4. Discussion


The SCUS has been described as a tropical upwelling system with low biological productivity compared to the mid-latitude

upwelling systems in the EBCS, due to the low nutrient content of the SUW waters that are the source of the water that

upwelled in the SCUS (Corredor, 1979). SUW waters form in the middle of the tropical Atlantic and enter the Caribbean at

~100 depth (Gordon, 1967), so they are not subject to significant biological processes that could consume or increase their

nutrient content, such as photosynthesis. Therefore, the main variations of the concentration of SUW nutrients before

reaching the upwelling system are almost exclusively due to the physical processes of mixing with adjacent water masses.

These processes may explain the observed decrease in salinity of the SUW waters that arise in the SCUS.



Turbulent diffusion processes are several orders of magnitude more intense than molecular diffusion processes (Thorpe,
2005b). In the central waters of the oceans, the most important turbulent diffusion processes are the mechanical mixing
generated by the horizontal shear of the currents and the salt fingers (McDougall and Ruddick, 1992). These two processes
have been observed coexisting in similar magnitudes ( $K_T \approx K_{sf} \approx 10^{-5}$ ) in the Canary upwelling system (east coast of the
North Atlantic), where mechanical mixing was more important on the surface, since under the thermocline, double diffusion
is the dominant process at the depth of the NACW waters (Arcos-Pulido et al., 2014). Our observations also show the
presence of these two diffusive processes in the SCUS, although the mechanical mixing is ~ 2 orders of magnitude more
intense (~ $10^{-3}$) in the whole water column and dominates the double diffusion even at the depth of the wNACW (~ 300 m).
The high mechanical diffusivity found in this paper was estimated indirectly from the density reversals (Thorpe scale) in the
water column density profile, since there are no direct measurements of turbulence. Although it has been reported that
indirect estimates are comparable with direct turbulence measurements (Park et al., 2014), additional direct measurements
are highly necessary to adjust current estimates of diffusion in the system.

Currents simulated by the Mercator model show the CaCU is flowing eastward along the continental shelf at the same depth
of the SSM (50-150 m depth), transporting SUW waters at 0, 4 m s$^{-1}$. This velocity is higher than that previously reported by
Andrade et al. (2003), possibly because their speed estimates were not based on direct measurements since they were based
on gesotrophic flow estimates from CTD data, which could underestimate the real speed of the CaCU. Unfortunately, there is
no way to determine the accuracy of the previous estimates and those reported in this study because there are no direct
measurements of the speed in the CaCU. The simulated currents also suggest a remote origin of the CaCU in front of the
Nicaragua shelf from a coastal branch of the western SUW waters which returns towards the east along the southern shelf of
the Caribbean. This controverts the superficial origin hypothesis in the PCG region previously proposed for the CaCU
(Andrade et al., 2003). Indeed, our CTD observations show the high thermal and saline stratification in the PCG region (Fig.
4c-d) makes subduction of surface waters is unlikely as a formation mechanism of the CaCU.

Our results show that CaCU provides the diluted SUW waters that are upwelled in the SCUS and, therefore, most of the
upwelled waters do not come directly from the central Caribbean. Instead it, the upwelled waters seems to come from the
PCG region located west of the SCUS. This region is a sub-basin of dilution due to the large inflow of fresh water by rain
and the discharge of large rivers in the Gulf of Darien (south-southwest Caribbean), which exceeds 2800 m$^3$ s$^{-1}$ (Beier et al.,
2017). The high coefficients of mechanical diffusivity observed in the surface suggest that the SUW waters transported at
subsurface by the CaCU could be mixed with the surface waters of low salinity in this region. This mixture is favored by the
predominant cyclonic circulation in the PCG region that generates an SSM elevation from 140 m to <100 m depth increasing
the contact of the SUW water with the surface water. The transportation of SUW by the CaCU determines an additional trip

of 2600 km and 4400 km that these waters must complete before reaching the upwelling zones in front of Colombia and Venezuela, respectively. Therefore, SUW waters will have an additional subsurface retention time of 2.5 (4.0) months during which they will be exposed to the vertical salt diffusion process, assuming a continuous CaCU velocity of $\sim 0.4$ m s$^{-1}$ as indicated by the Mercator modeled data. This longer residence time may be increasing (or decreasing) the concentration of

nutrients in the upwelled SUW waters that are supporting the biological productivity observed in the upwelling system.

There is an extensive observational evidence of double diffusion processes with the formation of salt fingers in the central waters of the tropical Atlantic and the Caribbean (Schmitt, 2005; Schmitt et al., 1987). Salt finger formation generates a turbulent salt transport $\sim 2$ orders of magnitude greater than molecular salt transport (Schmitt, 2005), which is comparable to

fluxes induced by mechanical turbulence or mesoscale eddies (Oschlies et al., 2003). SSince the main salt ions of seawater have molecular characteristics similar to the nutrients, the salt diffusivity is similar to the nutrient diffusivity (Hamilton et al., 1989). In the subtropical North Atlantic, the transport of nutrients towards the surface layer by the salt fingers has been observed $\sim 5$ times greater than the transport due to other turbulent processes in the thermocline, such as the generate by internal waves (Dietze et al., 2004). Thus, the diffusion of nitrate by salt fingers could account for ca. 20% of the supply of

new nitrogen in the tropical Atlantic (Fernández-Castro et al., 2015). The nutrients diffusion by salt fingers of has not been analyzed in the SCUS, although this information could be important to completely understand the balance of nutrients and the biological production observed in the system. In the Canary upwelling system located on the eastern edge of the North Atlantic, salt fingers generate an important vertical transport of nutrients comparable to other turbulent processes of vertical nutrient mixing (Arcos-Pulido et al., 2014). In the SCUS, the mechanical mixing exceeds the double diffusion mainly in the

surface, which can modify the concentration of nutrients of the SUW waters that eventually reach the surface in the upwelling system. Despite this, it is still unknown in the system if these processes contribute to enrich or decrease the concentration of nutrients in SUW waters. Further studies are required to quantify the variability of nutrient diffusive transport over time and between different sectors of the SCUS, its relationship with periods of intensification-weakening in the CaCU and its contribution to the biological productivity observed in this upwelling system.


## 5. Conclusions

Using CTD data from two different oceanographic campaigns (ANH and AMPs cruises), together with simulated data from the Mercator oceanic numeric model, the present study showed that the upwelled waters in the Southern Caribbean upwelling system come from modified SUW waters, which they have suffered a slight dilution of their salt content of $\sim 0.2$ g

kg$^{-1}$ in the PCG region. The simulated data shows that part of the SUW waters that enter through the passages between the Antilles and moves at subsurface along the Caribbean to the continental shelf of Nicaragua at the western edge of the basin, is returned to the east by the CaCU, a coastal undercurrent flowing along the edge of the platform at a speed of 4 m s$^{-1}$ below the main upwelling zones off Colombia and Venezuela. The CaCU is seasonally intensified with the intensification of the





coastal outcrop, providing most of the upwelled SUW waters. The transport by CaCU determines a longer retention time at
subsurface during which SUW waters undergone intense turbulent mixing processes ($> 10^{-3}$) that modify their salt content.
Most of the coastal SUW waters dilution is probably the effect of the mechanical mixing of SUW waters with low-salinity
surface waters in the PCG region, which occurs when these waters are transported at sub-surface under this region by the
CaCU. In addition to decreasing the salt content in SUW waters, these mixing processes can contribute to modify the
nutrient content in upwelling waters with still unknown effects on biological productivity..


## 6. Acknowledgments

This work was funded by the Autonomous National Fund Patrimony "Francisco José de Caldas" of the Administrative
Department of Science, Technology, and Innovation - COLCIENCIAS, through the grant No. FP44842-138-2016
COLCIENCIAS–INVEMAR, project code 210571451272. A. Rodriguez-Santana was funded by the FLUXES project
(CTM2015-69392-C3-3-R) of the Spanish National Research Program and European Regional Development Fund. Authors
also thank the financial support of the National Investment Projects Bank of Colombia - BPIN, at the Marine and Coastal
Research Institute - INVEMAR, project "Investigación científica hacia la generación de información y conocimiento de las
zonas marinas y costeras de interes de la nación" code 20170110000113.

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
