# Peer review of "Water masses and mixing processes in the Southern Caribbean upwelling system off Colombia"

_Ocean Science, 2019_

## Referee Comment (RC1) · Anonymous Referee #1 · 23 May 2019

This manuscript presents a study of the Caribbean upwelling system based on four cruises as well as the analysis of outputs from numerical simulations performed with the Mercator model. There are two main objectivesǍǎ: the first one is to determine the origin of the upwelled waters and the second one is to characterize mixing processes that may influence biological productivity. The upwelled waters, that are mainly consti- tuted of Subtropical Water Mass (SUW), are characterized by a local salinity maximum. This salinity maximum presents a strong seasonal variability and is significantly smaller than that of the SUW, as inferred from in-situ data. The pathway of these upwelled wa- ters is inferred from the Mercator modelǍǎoutputs: they originate from the Western Caribbean Sea and are transported by the intense Caribbean coastal undercurrent

(CaCU). Mixing processes are estimated and shown to be significant with vertical diffusivities of 10-4m2s-1 for double diffusive processes and 10-3m2s-1 for mechanical mixing. This mixing impacts the salinity distribution of these coastal upwelled waters but the impact on the nutrient content in the upwelling region is to be determined.

The manuscript is generally well written and presented with clear figures. The topic is interesting and the questions adressed relevant though I don't know to which extent the results are new, not being a specialist of the watermasses and circulation in the area. Also I find a lack of convincing results with respect to the points adressed.

My major concern is on mixing processes, with estimates of diffusion by salt fingering (SF), active below the subsurface salinity maximum, and by turbulence (T). The SF diffusvity is derived from a formula (eq. (3)) with out any reference. I guess that this formulation is derived from Schmitt (1981), in any case it should be mentionned. I wonder about the Âńˆˆbackground̆Âż constant value, which is large, as well as the maximum Ksf value, how were they prescribedˆĂ? The vertical eddy diffusivity, KT, is inferred from density overturns. The method is described with details except for the background value when no density overturns. Its application to in-situ data is not detailed. It would be interesting to know how the KT vertical sections shown in Figure 4 were obtainedˆĂ: indeed according to the N2 sections, the stratification is always stable, so one may wonder whether density overturns were resolved or not, how the computation for each individual vertical profile was performed (with or without a background value) and how the interpolation was perfomed for KT. It is also confusing to discuss the relative part of the salt fingering and turbulent mixing contribution to diffusive salt fluxes with a background Ksf taking into account other mixing processes (i.e. mechanically driven mixing, for instance by internal wave breaking). This analysis is not convincingˆĂ: this may result both from the lack of details provided and mostly from the inadequacy of the dataset to this aim.

In conclusion my advice would be to remove the part on mixing processes as it is not convincing(see details above). Also the objective is too ambitious owing to the data

available, without current measurements and microstructure measurements. Regarding the water mass part and upwelled waters pathways, it should be strenghtened with further analysis. For instance a lagrangian analysis based on retro-trajectories may be helpful to this purpose for tracing water masses pathways and provide more convincing results.

My recommandation is to submit a new manuscript focused on the circulation and water masses excluding mixing processes.

---

## Referee Comment (RC2) · Anonymous Referee #2 · 27 May 2019

Water masses and mixing processes in the Southern Caribbean upwelling system off Colombia by Marco Correa-Ramirez, Ángel Rodriguez-Santana, Constanza Ricaurte-Villota, and Jorge Paramo

Using more than 100 salinity and temperature profiles off the Colombian coast, the authors investigate the Southern Caribbean Upwelling System, particularly the path and role of the Subtropical Underwater. There is basic knowledge on the SCUS, but little is known on transports, mixing rates, fluxes, which all are of interest for the bio-geochemical impact of the SCUS, and could be used to compare to Eastern Boundary Upwelling Systems and to learn on upwelling systems in general. So the subject of

the manuscript is an interesting research question and suitable for this journal. The authors propose the Caribbean Coastal Undercurrent, which transports SUW, to be stronger than previously thought (a mean of 0.4m/s), and being the main feed of up-welling water to the SCUS. They deduce high diapycnal turbulent mixing rates through-out the water column in the SCUS (of order 10-3 m2/s), and conclude that the saline SUW gets modified by mixing with fresher waters on its way along the Western and Southern Caribbean Coast, before being upwelled.

Evaluating the manuscript, the data base seems not sufficient, spatially and in param-eters, to unravel the processes outside and inside the study region, which are the base of the authors' conclusions. For example, the concluded very high turbulent mixing rates are based only on Thorpe scale analysis of the CTD profiles, with unfortunately no ground truthing possible. The very high speed of the CaCU is based only on the Mercator model output. This hampers the ability of the data set to be the foundation for the ambitious goal to describe water mass changes, transport paths, mixing impact, particularly when the findings are unexpected like here (unexpected strong undercur-rent, unexpected strong diapycnal mixing and salt flux, unexpected path of the SUW feeding the upwelling). Missing to support the proposed scenario is data on salin-ity and mixing in the PCG, how to consolidate mixing and salinity development along the CaCU/SUW off the Colombian coast, evidence that the pathway of SUW from the Central Caribbean towards upwelling is negligible compared to the CaCU pathway. Besides, there are several critical points, and some method aspects that need to be shown (see below). However, I see possibilities if focussing the paper on the SUW salinity balance in the region of the observational data base. Estimates of current ve-locity and vertical mixing (after a rough calculation based on 0.2m/s CaCU speed and Fig.4 values) support a salinity loss of 0.1 or 0.2 by vertical mixing along the path of the CaCU off Colombia. Together with estimates of CaCU transport and Ekman trans-port, some plausible value range for horizontal mixing, and given consistency with the salinity field, a salinity budget could lead to an estimate which pathway contributes how much to the upwelling. Given that such information can be inferred for the SCUS,

the paper could also seek answers what is different to EBUS in dynamical processes affecting biogeochemistry (and why).

This would however be a considerable change of focus ('The subsurface salinity maximum in the Southern Caribbean upwelling system off Colombia'?) and considerable additional work, so that I recommend not to give major revision, but reject and encourage submission of a refocused study. I would appreciate if the existing observational data could be published and used, particularly if the strong mixing can be verified.

Specific comments:

Given the high turbulent mixing inferred is real, salt finger influence seems negligible, therefore I would recommend to give it less space in the paper, as it has very minor impact on mixing and salt fluxes (<10-2 g/kg m/d for the lower boundary of the SUW, not quantified for the upper boundary, Fig.3).

The inferred high turbulent mixing is entirely based on the Thorpe scale analysis. As there is no ground truthing possible, and the reported K = 10-3 throughout the water column seems very high to me, it seems necessary to give the Thorpe scale analysis a more prominent space in the paper. Make used CTD data accessible/visible, show profiles, show that instabilities and overturns are discernible above sensor noise and salinity spikes, show some estimate of uncertainty based on the CTD sampling rate, the lowering velocity, the postprocessing of CTD data. With the high stratification of N2 above 10-4, LT of 0.2m would have to be discerned with confidence from 0.6m, in order to discern K = 10-4 from 10-3. At a lowering rate of 1m/s this usually means 5 to 15 raw values of the CTD sensors along a typical overturn. For my feeling, this makes Thorpe scale analysis hard under these stratification conditions. This should clearly be addressed, and shown that issues can be solved by the chosen processing, in order to convincingly lay down the high diapycnal mixing rate is no artefact.

I would recommend not to speculate about the impact on nutrient fluxes, as there is no information on the nutrient field at all. Maybe a sentence in the Discussion, what
nutrient fluxes to the mixed layer a diffusion coefficient of K = 10-3 would cause in typical oligotrophic ocean regions (there should be some information around on nutrient gradients below the mixed layer for the tropical Atlantic and the Caribbean).

The salinity maps seem to contradict the authors' conclusion that the fresher CaCU feeds the upwelling, and that there's no direct feed from the North: Fig.7 shows increasing salinity along the path of the CaCU, particularly when the CaCU is strong in January to March, and as the map follows the vertical salinity maximum, this increasing salinity cannot be caused by vertical mixing.

Looking at the model currents (Figs.6 and 8), it seems the CaCU rather has its maximum at 0.4m/s, not its average as is stated in the abstract and other places. From Fig.8 I would maybe read an average of about 0.2m/s (0.1m/s at Guajira), which would then also fit the observed current velocities in Andrade et al. (2003), who show current sections with core velocities of up to 0.4m/s (their Fig.3) and lower currents at Guajira (their Fig.4).

Lines 90-109: Plase clarify what CTD was used on AMP cruises. What were conditions, lowering rates, processing, accuracies? (Seems particularly important for the CTD data role in Thorpe scale analysis)

L140: There is a discrepancy between Ksf defined in equation 3 and respective values shown in Fig 3. (Equation 3 as it is would not allow Ksf<3*10-5). If Ksf is meant to be the surplus mixing effect of salt fingering, Kinf should probably not be in equation 3.

L149ff: Please specify and show how you applied the data processing (threshold values chosen, identification of overturns...), and verified that the chosen processing works successfully.

L185-187: Please show evidence/references/analysis of defined regions, to support the reported 0.2 difference. The search for reasons is good for the discussion.

L223-224: N2 alone is not sufficient to evaluate the susceptibility to shear instability.

L235: Flux is not inherently dependent on total exposure time to mixing. Perhaps you had in mind the salinity budget of SUW, and refer to the total salt transport off SUW during the time it is following the CaCU. This budget is in fact worth looking at, I believe (see general comments above).

L246: Fig.5 (model based sections of S and T) should be rethought. In the present form of presentation it allows no real comparison to observations (maybe do a difference plot?), and it allows no conclusion how realistic the model circulation may be (maybe compare the density fields?)

L254: What does MSE mean (also later in the text)?

L262-268: Please specify if Fig.6 is for a single model year or a climatological seasonal cycle.

L270: probably 'eastward' is meant

L288-290: There is a discrepancy here between the 'could suggest' for some of the main points of the paper, and the certainty with which these conclusions are stated in L418 and in the Conclusion.

L333-337: The last paragraph of subsection 3.2 contains current velocity values of maybe a factor 10 too high, compared to the model output they refer to. In conclusion as well.

L376: The T-S-Diagram shows high variability in SUW salinity, between 36.8 and 37.1. From previous figures I could not see that it is a reduction of salinity from West to East, but it rather seems to show that SUW in the upwelling is variable between 'diluted' and 'Central Caribbean' SUW, and contains all mixtures of the two.

L426: Please support the large distances for the CaCU path. From the maps it seems like 1000 to 1500km in total. Months of the reported salt flux from SUW to adjacent waters at a mixing rate of 10-3 m2/s would mean several psu salinity change.

---

## Author Comment (AC1) · 29 Jun 2019

Using more than 100 salinity and temperature profiles off the Colombian coast, the authors investigate the Southern Caribbean Upwelling System, particularly the path and role of the Subtropical Underwater. There is basic knowledge on the SCUS, but little is known on transports, mixing rates, fluxes, which all are of interest for the biogeochemical impact of the SCUS, and could be used to compare to Eastern Boundary Upwelling Systems and to learn on upwelling systems in general. So the subject of the manuscript is an interesting research question and suitable for this journal. The authors propose the Caribbean Coastal Undercurrent, which transports SUW, to be

stronger than previously thought (a mean of 0.4m/s), and being the main feed of upwelling water to the SCUS. They deduce high diapycnal turbulent mixing rates throughout the water column in the SCUS (of order 10-3 m2/s), and conclude that the saline SUW gets modified by mixing with fresher waters on its way along the Western and Southern Caribbean Coast, before being upwelled.

Evaluating the manuscript, the data base seems not sufficient, spatially and in parameters, to unravel the processes outside and inside the study region, which are the base of the authors' conclusions. For example, the concluded very high turbulent mixing rates are based only on Thorpe scale analysis of the CTD profiles, with unfortunately no ground truthing possible. RR:// This is explained in detail in the specific comments below.

The very high speed of the CaCU is based only on the Mercator model output. This hampers the ability of the data set to be the foundation for the ambitious goal to describe water mass changes, transport paths, mixing impact, particularly when the findings are unexpected like here (unexpected strong undercurrent, unexpected strong diapycnal mixing and salt flux, unexpected path of the SUW feeding the upwelling). Missing to support the proposed scenario is data on salinity and mixing in the PCG, how to consolidate mixing and salinity development along the CaCU/SUW off the Colombian coast, evidence that the pathway of SUW from the Central Caribbean towards upwelling is negligible compared to the CaCU pathway. R:// This is explained in detail in the specific comments below.

Besides, there are several critical points, and some method aspects that need to be shown (see below). However, I see possibilities if focussing the paper on the SUW salinity balance in the region of the observational data base. Estimates of current velocity and vertical mixing (after a rough calculation based on 0.2m/s CaCU speed and Fig.4 values) support a salinity loss of 0.1 or 0.2 by vertical mixing along the path of the CaCU off Colombia. Together with estimates of CaCU transport and Ekman transport, some plausible value range for horizontal mixing, and given consistency with

the salinity field, a salinity budget could lead to an estimate which pathway contributes how much to the upwelling. Given that such information can be inferred for the SCUS, the paper could also seek answers what is different to EBUS in dynamical processes affecting biogeochemistry (and why). R:// This is explained in detail in the specific comments below.

This would however be a considerable change of focus ('The subsurface salinity maximum in the Southern Caribbean upwelling system off Colombia'?) and considerable additional work, so that I recommend not to give major revision, but reject and encourage submission of a refocused study. I would appreciate if the existing observational data could be published and used, particularly if the strong mixing can be verified.

Specific comments: Given the high turbulent mixing inferred is real, salt finger influence seems negligible, therefore I would recommend to give it less space in the paper, as it has very minor impact on mixing and salt fluxes (<10-2 g/kg m/d for the lower boundary of the SUW, not quantified for the upper boundary, Fig.3). R:// The Kt estimations based in the Thorpe escale were revised in detail. In this task we found a code error in the Brunt Vaisalla calculus that biased our Kt estimates. Solving this mistake, the Kt corrected estimates are now about 10-5 s-1, which is comparable with the Ksf magnitudes. With this corrected values, our results suggest both processes are important in the salt flux, although each one od them domain a different depth ranges: the salt fingers are more important in the salt flux towards depth and the mechanical mixing contributes to the salt flux towards surface. These findings is now included in the revised version of the manuscript.

The inferred high turbulent mixing is entirely based on the Thorpe scale analysis. As there is no ground truthing possible, and the reported K = 10-3 throughout the water column seems very high to me, it seems necessary to give the Thorpe scale analysis a more prominent space in the paper. R:// As explained above, the Kt estimates were revised and corrected, producing values in the expected order of magnitude of 10-5. We expanded the Methods section that includes a detailed explanation of

Thorpe's analysis in L174-197. An on-ground verification of the Kt estimates is beyond the scope of the present study and, throughout the manuscript, we suggest the need for additional studies with direct measurements of turbulence, such as those carried out with micro profilers, to contrast the current estimates. However, the Thorpe scale is currently a standard procedure that has been used in more than 15 articles published in the last 10 years.

Make used CTD data accessible/visible, show profiles, show that instabilities and overturns are discernible above sensor noise and salinity spikes, show some estimate of uncertainty based on the CTD sampling rate, the lowering velocity, the postprocessing of CTD data. R:// We include a new figure (new figure 2) that shows step by step how the calculation of Kt was performed. This figure shows how the overturns were estimated and which were not considered for the Kt estimate because they probably corresponded to artifacts caused by salinity peaks. In L125-126 of the revised manuscript was include information about the the CTD sampling rate (1 Hz) and the lowering velocity (âĹij 0.25 m s-1) used in AHN and AMP cruises. Besides, the Lines 115-124 contain a new detailed explanation of the processing applied to the CTD data before the coefficient diffusivity estimations. Here we also introduce a new methodology to removes salinity spikes that were used in the revised manuscript.

With the high stratification of N2 above 10-4, LT of 0.2m would have to be discerned with confidence from 0.6m, in order to discern K = 10-4 from 10-3. At a lowering rate of 1m/s this usually means 5 to 15 raw values of the CTD sensors along a typical overturn. For my feeling, this makes Thorpe scale analysis hard under these stratification conditions. This should clearly be addressed, and shown that issues can be solved by the chosen processing, in order to convincingly lay down the high diapycnal mixing rate is no artefact. R:// as explained above, high Kt difisivities reported in the previous version were overestimated due a code error in the Brunt Vaisalla estimation. This error was corrected and new estimations in the revised manuscript are about 10-5 s-1 as expected.

I would recommend not to speculate about the impact on nutrient fluxes, as there is no information on the nutrient field at all. Maybe a sentence in the Discussion, what nutrient fluxes to the mixed layer a diffusion coefficient of K = 10-3 would cause in typical oligotrophic ocean regions (there should be some information around on nutrient gradients below the mixed layer for the tropical Atlantic and the Caribbean). R:// Nutrients sources and fates are remain open questions in upwelling systems. For us is a natural next step to extrapolate our findings with the perspective of the nutrients cycling. However we agree with the reviewer and restrict the discussion of nutrients to the Discussion section.

The salinity maps seem to contradict the authors' conclusion that the fresher CaCU feeds the upwelling, and that there's no direct feed from the North: Fig.7 shows in creasing salinity along the path of the CaCU, particularly when the CaCU is strong in January to March, and as the map follows the vertical salinity maximum, this increasing salinity cannot be caused by vertical mixing. R:// The salinity map in fig. 7 (Fig. 8 in revised version) showed CaCU has its lower salinity in the PCG region. We postulates this could be caused by a major salt loos in this dilution sub-basin. With the upwelling intensification, the diluted SUW are transported by an intensified CaCU and introduce just in the Colombian Guajira shelf. But during the upwelling season the CaCU is not intense off Venezuela (as explained in L381-388 and showed in Fg. 9) and high salinity waters remain in this region, causing this apparent zonal increasing in salinity suggested by the reviewer. This apparent gradient disappear during the upwelling relaxation when the CaCU is intensified off Venezuela introducing more diluted SUW.

Looking at the model currents (Figs.6 and 8), it seems the CaCU rather has its maximum at 0.4m/s, not its average as is stated in the abstract and other places. From Fig.8 I would maybe read an average of about 0.2m/s (0.1m/s at Guajira), which would then also fit the observed current velocities in Andrade et al. (2003), who show current sections with core velocities of up to 0.4m/s (their Fig.3) and lower currents at Guajira (their Fig.4). R:// In fact the reported 0,4 m s-1 value in the previous version correspond

to the maximum speed. We includes a new subplot in the figure 7 (fig. 8 in the revised version) with the Three-year climatological average (2016-2018) of the currents velocity in the core of the CaCU. The mean value estimated of the 2,8 m s-1 was changed along the revised manuscript.

Lines 90-109: Plase clarify what CTD was used on AMP cruises. What were conditions, lowering rates, processing, accuracies? (Seems particularly important for the CTD data role in Thorpe scale analysis) R:// Done. This information was included in L105-108 and L125-126 of the revised manuscript.

L140: There is a discrepancy between Ksf defined in equation 3 and respective values shown in Fig 3. (Equation 3 as it is would not allow Ksf<3*10-5). If Ksf is meant to be the surplus mixing effect of salt fingering, Kinf should probably not be in equation 3. R:// Part of the Ksf background values correspond to the Kinf = 3x10-5, which is a constant value that account for the diapycnal diffusion due to processes unrelated to double diffusion, like the internal wave breaking (Schmitt, 1981). This value should be considered when estimating the total diffusivity of the salt, since Kinf difusivities are no related with salt fingers neither the shear instabilities generating the mechanical mixing. To ensure a better comparison between Ksf and Kt, we decided do not add this constant value in the Ksf estimates of the revised manuscript, which are shown in the new figures 2 and 4. This change are now explained in L171-172 "This coefficient was not considered in the Ksf estimates to guarantee a direct comparison only between the double diffusion process and mechanical diffusion due to shear instabilities."

L149ff: Please specify and show how you applied the data processing (threshold values chosen, identification of overturns...), and verified that the chosen processing works successfully. R:// Done. The explanation of the eddy diffusivity estimation (KT) thought Thorpe scale was expanded in the revised manuscript in L174-196. Questionable Thorpe scales were detected and removed from using an overturn verification trough the overturn ratio Ro (equation 2). Since values of Ro<0.25 suggest non-symmetric density overturns that could be caused by residual salinity spiking (Park et

al., 2014), the Thorpe scale associated with these Ro values are excluded.

L185-187: Please show evidence/references/analysis of defined regions, to support the reported 0.2 difference. The search for reasons is good for the discussion. R:// Done. We include a new panel in figure 3a showing several Argo floats profile from the central Caribbean that were done during the same years of ANH cruises. These profiles showing the salinity at the SSM depth is between 0.05 and 0.14 g kg-1 higher than the observed off the Colombian shelf in the dry an rainy seasons respectively. This adjusted difference is included in L223-226 of the revised manuscript.

L223-224: N2 alone is not sufficient to evaluate the susceptibility to shear instability. R:// We agree with the reviewer. With this paragraph we just wanted to point out that N2 is opposed to the formation of instabilities and overturns. For more clarity we change this sentence to: "The mechanical turbulence produced by the vertical shear of the horizontal currents is another process that can potentially contribute to the mixing of the SUW waters with the adjacent water masses above and below the depth of the SSM. Since the stratification in the water column generally opposites the occurrence of shear turbulence, stratification also provides information on the susceptibility to mechanical mixing.". This paragraph is found in L267-270 of the revised manuscript.

L235: Flux is not inherently dependent on total exposure time to mixing. Perhaps you had in mind the salinity budget of SUW, and refer to the total salt transport off SUW during the time it is following the CaCU. This budget is in fact worth looking at, I believe (see general comments above). R:// Salt balance in CaCu is in fact an important issue to determine, but is beyond of the scope of the present study. In the new version of Figure 6 we showed that the modeled data available does not accurately reproduce the salt and the temperature at the SSM depth and, because that, this data is not useful to this objective. Besides, there are few observatonal data to accomplish this task. We believe that further studies including tracer experiments and micorstructure profiles, are required to perform an an accurate balance.

[Figure]

L246: Fig.5 (model based sections of S and T) should be rethought. In the present form of presentation it allows no real comparison to observations (maybe do a difference plot?), and it allows no conclusion how realistic the model circulation may be (maybe compare the density fields?) R:// We change the fig. 5 (figure 6 in the revised manuscript) with a new one including the average and deviations profiles of the ANH-I and Mercator profiles. This new figure also includes the difference between the observed an modeled profiles. This figure shows modeled profiles underestimates the $\Theta$ (in $\sim 2^\circ$C) and SA (in $\sim 0.4$ g kg-1) mainly between 150-250 m depth, where the main termocline and halocline are located (L294-307).

L254: What does MSE mean (also later in the text)? R:// This was an erroneous acronym of the SSM, that was already corrected in the revised manuscript.

L262-268: Please specify if Fig.6 is for a single model year or a climatological seasonal. R:// Figures 6,7 and 8 (7,8 and 9 in the revised manuscript) were redone using a 3-year climatological mean of Mercator modeled outputs. This is now explained in their respective figure captions.

L270: probably 'eastward' is meant R:// Corrected.

L288-290: There is a discrepancy here between the 'could suggest' for some of the main points of the paper, and the certainty with which these conclusions are stated in L418 and in the Conclusion. R:// We revised and change these sentences. We agree with the reviewer in the fact of our results suggest the dilution of the SUW but are not entirely conclusives. We do not exludes the needed of more studies in the zone to veify these results.

L333-337: The last paragraph of subsection 3.2 contains current velocity values of maybe a factor 10 too high, compared to the model output they refer to. In conclusion as well. R:// This was an typo error. I was corrected with the velocityy values about 0.2 m s-1

L376: The T-S-Diagram shows high variability in SUW salinity, between 36.8 and 37.1. From previous figures I could not see that it is a reduction of salinity from West to East, but it rather seems to show that SUW in the upwelling is variable between 'diluted' and 'Central Caribbean' SUW, and contains all mixtures of the two. R:// The T-S diagram of the AMP cruises was included because the SUW in this figure showed the same salinity reduction as observed in the ANH cruises. If the SUW waters of the central Caribbean could enter the Guajira platform, the TS plots should be similar to the Argo profiles shown in Figure 3b, but this is not the case. We found no evidence of the presence of SUW in the central Caribbean, as suggested by the reviewer.

L426: Please support the large distances for the CaCU path. From the maps it seems like 1000 to 1500km in total. Months of the reported salt flux from SUW to adjacent waters at a mixing rate of 10-3 m2/s would mean several psu salinity change. R:// The distances from the Nicaragua shelf to the upwelling zones front Colombia and Venezuela are 1300 and 2200 km, respectively. But the not diluted SUW that are only about 150 north from these areas, must travel before towards the Nicaragua shelf and return back to the upwelling zones, traveling twice this distance. This is now expanded in L482-486 of revised manuscript.

---

## Author Comment (AC2) · 29 Jun 2019

Anonymous Referee #1 This manuscript presents a study of the Caribbean upwelling system based on four cruises as well as the analysis of outputs from numerical simulations performed with the Mercator model. There are two main objectives: the first one is to determine the origin of the upwelled waters and the second one is to characterize mixing processes that may influence biological productivity. The upwelled waters, that are mainly constituted of Subtropical Water Mass (SUW), are characterized by a local salinity maximum. This salinity maximum presents a strong seasonal variability and is significantly smaller than that of the SUW, as inferred from in-situ data. The pathway

of these upwelled waters is inferred from the Mercator model outputs: they originate from the Western Caribbean Sea and are transported by the intense Caribbean coastal undercurrent (CaCU). Mixing processes are estimated and shown to be significant with vertical diffusivities of 10-4m2s-1 for double diffusive processes and 10-3m2s-1 for mechanical mixing. This mixing impacts the salinity distribution of these coastal upwelled waters but the impact on the nutrient content in the upwelling region is to be determined.

The manuscript is generally well written and presented with clear figures. The topic is interesting and the questions adressed relevant though I don't know to which extent the results are new, not being a specialist of the watermasses and circulation in the area. Also I find a lack of convincing results with respect to the points addressed. My major concern is on mixing processes, with estimates of diffusion by salt fingering (SF), active below the subsurface salinity maximum, and by turbulence (T). The SF diffusvity is derived from a formula (eq. (3)) with out any reference. I guess that this formulation is derived from Schmitt (1981), in any case it should be mentionned. R:// In fact the Salt finger difusivities are derived from Schmitt approximation. The The missing reference was incorporated in line 169.

I wonder about the background constant value, which is large, as well as the maximum Ksf value, how were they prescribed? R:// Part of the Ksf background values correspond to the Kinf = 3x10-5, which is a constant value that account for the diapycnal diffusion due to processes unrelated to double diffusion, like the internal wave breaking (Schmitt, 1981). This value should be considered when estimating the total diffusivity of the salt, since Kinf difusivities are no related with salt fingers neither the shear instabilities generating the mechanical mixing. To ensure a better comparison between Ksf and Kt, we decided to follow the suggestion of the reviewer and not add this constant value in the Ksf estimates of the revised manuscript, which are shown in the new figures 2 and 4. This change are now explained in L171-172 "This coefficient was not considered in the Ksf estimates to guarantee a direct comparison only between the

double diffusion process and mechanical diffusion due to shear instabilities."

The vertical eddy diffusivity, KT, is inferred from density overturns. The method is described with details except for the background value when no density overturns. R:// The explanation of the eddy diffusivity estimation (KT) thought Thorpe scale was expanded in the revised manuscript (L174-196). The profile sections where no overturns were detected do not necessarily mean there is no vertical mixing, because small overturns lower than the vertical CTD sampling intervals could exist but not be detected. Considering the smallest detectable overturn, a conservative value of $1 \times 10^{-6}$ m2 s-1 was set in this regions, as suggested by Zhu and Zhang (2018). This is now explained in L190-194.

Its application to in-situ data is not detailed. It would be interesting to know how the KT vertical sections shown in Figure 4 were obtained: indeed according to the N2 sections, the stratification is always stable, so one may wonder whether density overturns were resolved or not, how the computation for each individual vertical profile was performed (with or without a background value) and how the interpolation was performed for KT. R:// We includes a new figure (figure 2) showing an example of how the Kt computation was done. The N2 profiles showed are calculated from smoothed density profiles with a 10 m mobile mean, to avoid that the artifacts caused by density spikes biasing the Kt estimates, as now explained in L190. Because that, small stratification changes due overturns are not evident in N2 profiles. Besides, the diffusivities below the 300 m depth where the respective N2 values were lower than $3 \times 10^{-5}$ s-2 are excluded since low N2 values could produce high erroneous diffusivities.

For all sections showed in the manuscript figures we use DIVA interpolation method, as explained in L35-39 of Methodology.

It is also confusing to discuss the relative part of the salt fingering and turbulent mixing contribution to diffusive salt fluxes with a background Ksf taking into account other mixing processes (i.e. mechanically driven mixing, for instance by internal wave breaking).

This analysis is not convincing: this may result both from the lack of details provided and mostly from the inadequacy of the dataset to this aim. R:// As explained in the above responses, we decide do not consider Kinf in the recalculated Ksf estimates of the revised manuscript, following the reviewer suggestion. With this change is now possible a direct comparison between Ksf and Kt diffusivities.

In conclusion my advice would be to remove the part on mixing processes as it is not convincing(see details above). Also the objective is too ambitious owing to the data available, without current measurements and microstructure measurements. Regarding the water mass part and upwelled waters pathways, it should be strenghtened with further analysis. For instance a lagrangian analysis based on retro-trajectories may be helpful to this purpose for tracing water masses pathways and provide more convincing results. R:// We improved Figure 8 (Figure 9 in the revised version) with the inclusion of streamlines to observe the flow path of subsurface waters in the Caribbean. We also show in the new version of Figure 6 that the modeled data available does not accurately reproduce the salt and the temperature at depth. Because of that, the use of this data to track water masses, as the reviewer suggests, would not produce a precise result. We believe that adequate tracking of water masses should be carried out through the use of tracer experiments in further studies.

My recommandation is to submit a new manuscript focused on the circulation and water masses excluding mixing processes. R:// Diffusivity and salt flux estimates have not be done previously in the region. We considered important report this estimations as a valuable piece of information needed to contributes of a better understanding of this upwelling system. Besides, is an important finding establish the depth where the different mixing processes domain the proprieties transport in the water column of this upwelling system.